# Autonomous Trajectory Planning for Spray Painting on Complex Surfaces Based on a Point Cloud Model

**DOI:** 10.3390/s23249634

**Published:** 2023-12-05

**Authors:** Saul Nieto Bastida, Chyi-Yeu Lin

**Affiliations:** 1Department of Mechanical Engineering, National Taiwan University of Science and Technology, Taipei 106, Taiwan; d10903814@mail.ntust.edu.tw; 2Center for Intelligent Manufacturing Innovation, National Taiwan University of Science and Technology, Taipei 106, Taiwan

**Keywords:** autonomous path planning, free-form surface, point cloud based, spray painting

## Abstract

Using teach pendants or offline programming methods can generate tool paths for robot manipulators to carry out production activities, such as spray painting on objects of different geometries. This task, in which the complexity of painting the surface is one of the main challenges, requires highly skilled operators. In addition, the time spent setting up a robot task can be justified for the mass production of the same workpiece. However, it is inconvenient for low-production and high-variation production lines. In order to overcome these challenges, this study presents an algorithm to autonomously generate robot trajectories for a spray-painting process applied to objects with complex surfaces based on input 3D point cloud data. A predefined spherical mesh wraps the object, organizing the geometrical attributes into a structured data set. Subsequently, the region of interest is extracted and isolated from the model, which serves as the basis for the automatic path-planning operation. A user-friendly graphical user interface (GUI) is developed to define input parameters, visualize the point cloud model and the generated trajectory, simulate paint quality using a color map, and ultimately generate the robot’s code. A 3D sensor is used to localize the pose of the workpiece ahead of the robot and adjust the robot’s trajectory. The efficacy of the proposed approach is validated first by using various workpieces within a simulated environment and second by employing a real robot to execute the motion task.

## 1. Introduction

The popularization of autonomous production technologies in a wide variety of applications includes robotic spray painting, and researchers are working toward improving this task in terms of efficiency and quality. The manual teaching method is the most commonly used method for creating trajectories for industrial robots. This task is time consuming and relies mainly on the programmer’s technical expertise. Due to the complexity of an object’s geometry, it can be challenging to accurately define the trajectory of the spray gun over all the surfaces of the workpiece, affecting the quality of the final coating. The time-consuming manual teaching method offers a suitable solution for mass production, but a more efficient method is desirable for high-variety and small production lines.

During manual spraying, the paint is scattered into the surrounding environment, causing chemical pollution and posing a severe risk to the health of the workers [1]. Spraying operations have been found to have high rates of reportable incidents by the Occupational Safety and Health Administration (OSHA) due to the nature of being an ergonomically difficult job requiring the manipulation and fine control of tools for prolonged periods [2]. Automatic systems have been developed [3] to solve the problems caused by traditional manual spraying. With significant advantages such as quality control, waste reduction, repeatability, and faster cycle times, automated painting is a viable solution to this problem. These innovative elements inspired the idea of the autonomous robotic painting system described in this research article.

Several techniques have been given in autonomous path planning and paint-deposition simulations. Previous studies in autonomous path planning compared the benefits and drawbacks of the existing approaches, most of which are based on tessellated representations in the “Standard Tessellation Language” (STL) format or 3D CAD models [4]. The authors of [1] provide a more recent review of actual tool-path-planning approaches for spray painting in which a point cloud is included as a geometrical representation of the model. Another comparison was made in [5], which includes parametric models and tessellated models as geometry representations but also considers how the model is obtained; in this case, two methods were mentioned: a CAD-based method in which geometries are generated using any existing CAD software and a sensor-based method which includes the use of sensors such as depth cameras, light detection and ranging (Li-DAR), and coordinate-measuring machines. In [6], an offline automatic trajectory generation approach based on free-form tessellated surfaces is presented. Still, it is limited to planar and non-highly-curved surfaces considered a single patch. A similar procedure called the “mesh following technique” was presented in [7] which is focused on generating tool paths based on tessellated surfaces presenting holes and irregular edges. The authors of [8] propose a method based on tessellated surfaces to create a total coverage trajectory using a grid projected on the surface; however, it is limited to basic shapes, and the part must be considered a single patch. Some advantages and disadvantages of using tessellated surfaces listed by the authors of [4] are provided below.

Advantages:It is a simple surface representation method;The area and the normal vector of each small triangle are known and can be used to represent the paint distribution on the entire surface;The triangulated surface is considered a single object.

Disadvantages:No geometrical information like edges;Mesh data are unorganized and more likely to present errors like overlapping facets or data loss during surface reconstruction.

The size of the data set tends to be large and difficult to process. On the other hand, automation systems with high product volumes frequently use CAD-based robot path planning. It is desirable to create a complete 3D model of the robotic cell. Before creating a program ready for robot operation, the user can test the reachability of the robot’s movements and simulate the process [9]. Given the convenience of using 3D CAD model features (face, wire, edges, etc.), the task of defining and generating a path to perform a variety of industrial tasks (glue dispensing, welding, inspection, painting, etc.) is less complicated than using STL files or a point cloud, which is based on an unorganized set of points and edges. Some commercial software, such as Robotmaster^®^ 2023 (Saint-Laurent, QC, Canada), Visual Components Robotics OLP 4.8.0 (Espoo, Finland), and Dürr^®^ DXQ3D.onsite process simulation tool (Stuttgart, Germany), offers advanced CAD-model-based path-planning functions specifically for spray-painting tasks in which it is only necessary to input some spray parameters to obtain a robot trajectory and a paint-deposition simulation as outputs. Other approaches, as in [10,11] in which an OLP platform based on Open CASCADE Technology (OCCT) [12] was presented, used CAD information to create a 3D scan and a glue application trajectory and to generate a robot program. Reference [13] presents a form of CAD-based trajectory optimization, but an initial trajectory must be defined as an input before the algorithm can optimize the point position to achieve the desired paint thickness quality on the surface. The algorithm works efficiently on low-complexity surfaces but has not been demonstrated on high-complexity surfaces. An autonomous CAD-based robot-path-planning system has been created for spray painting on compound surfaces, common in the automotive manufacturing industry [14]. The area-weighted average of larger patches generated by stitching together small surfaces is used in this method to estimate the spray gun’s orientation. Some advantages and disadvantages of using 3D CAD models [4] are listed below.

Advantages:Achieving an accurate illustration of a component’s geometry, including solids, surfaces, splines, arcs, lines, points, and other features;Information about the material is included.

Drawbacks:Every feature is independent. The more complex the object, the more individual its features, making analysis more difficult;If the paint quality is to be evaluated, a mesh representation must be generated.

Other related works, such as [15], describe vision-based measurement devices that determine the target points of the spray gun by measuring the relative posture between the painting end effector and the surface. Although studies on CAD- or mesh-based tool path planning for controlling robot arm motion have been conducted, techniques for complex spraying tasks employing 3D point cloud model data are still mostly unexplored. The authors of [16] described a point-cloud-based tool-path-planning method; however, the path-planning algorithm works only on simple surfaces with rectangular edges and edges aligned with the x and y axes. A slicing method in point clouds was presented in [17] which is suitable for simple surfaces and functions by intersecting parallel planes in a determined direction. It depends on properly selecting a piece’s orientation to define the slicing plane orientation and the corresponding path points. In a recent research study, machine-learning techniques were explored [18], and five data-driven conceptual approaches that have the potential to improve the efficiency of solving trajectory planning by taking advantage of existing data sets to generate new tasks based on their similarity to previous ones were proposed.

Industries have experienced a rise in demand for effective spraying applications on workpieces of various complex surfaces, which calls for more robust and effective methods for quantifying geometric data and path-planning techniques. Most shape recognition algorithms rely on geometrical features or shape boundary information. Reconstructing scanned geometries as they would have originally been built using CAD processes, i.e., as lofted cross sections along a path, remains a significant challenge in digital shape reconstruction.

## 2. System Overview

This work aims to create an autonomous path-planning system for advanced spraying jobs on workpieces with complex surface geometries, employing either a point cloud or a CAD model of the work piece, and a 6-DOF manipulator as the painting robot. In this work, a novel method for trajectory planning in spray-painting applications is proposed, where a predefined spherical mesh grid is used to wrap a point cloud and approximate the surface geometry in an organized set of points. The use of a variable paint flux is also proposed to improve the paint thickness quality based on paint thickness simulation and constant spray-gun velocity optimization.

In order to accomplish complete paint coverage of the section of interest, the first step involves using a sampling algorithm to obtain a point cloud from a 3D CAD model or directly from a 3D scanning system. In the second step, the parameters for spray painting are defined, along with a mathematical model of paint deposition on the surface. This is followed by the third step, in which the proposed path planning algorithm extracts geometrical information from the point cloud and generates a raster pattern. In the fourth step, the robot velocity and spray-paint flux are calculated to guarantee that the paint film thickness stays within a predetermined range. Finally, the obtained trajectory and the parameters are converted into a robot program to carry out the task in a production line, as depicted in Figure 1.

The main contributions of the work presented in this paper to address the challenges of autonomous trajectory planning can be described as follows:The trajectory is generated using only a point cloud as the workpiece’s geometrical information. The point cloud can be generated through 3D scanning or generated from a CAD model;An algorithm for generating spray-painting trajectories autonomously that achieve full coverage of a complex surface while keeping the paint film thickness within a given range;A trajectory-generation algorithm capable of addressing the challenges presented by complex surfaces like complex shapes, holes, cavities, irregular edges, and high roughness;The ability to handle workpieces in non-predefined orientations without the use of special fixtures or established positions;A simulation of paint thickness to validate the generated trajectory in a virtual environment and use this information to improve the spraying parameters as the paint flux.

In more detail, Section 3 presents a generalized methodology to automate the generation of spray-painting paths and 3D surface reconstruction based on 3D point cloud data; Section 4 provides the mathematical model used to calculate the paint deposition on free form surfaces; Section 5 describes the proposed path planning algorithm based on “wrapping” the point cloud to extract the surface of interest; Section 6 presents the robot end effector velocity calculation approach used on this research; and Section 7 details the variable paint flux calculation to improve the uniformity on the paint film, of which the proposed algorithm was evaluated in a simulated environment to validate its efficacy and efficiency. Several case studies with different workpiece shapes are discussed in Section 8 to illustrate the potential to use our approach to create paths for spraying tasks, followed by a conclusion of the research work completed for this article in Section 9.

## 3. Materials and Methods

Autonomous tasks using computer vision and robotic systems can be described as a Sense–think–act process. In the case of autonomous spray painting, the “sensing” section belongs to the 3D scanning process where the part’s geometry and environment configuration are obtained; a computer performs the “think” section by applying specific algorithms to extract the geometrical information of the part; and finally, the spraying paths and associated robot trajectory information for the robot are generated to perform the painting “action”.

Based on a 3D point cloud model representing a workpiece, a spray-painting path is generated to achieve total coverage of a 3D part with a complex shape. The point cloud is obtained from a 3D scanning system or applying a sampling algorithm to a CAD model, then preprocessed to prepare the data needed. The next step is to extract 3D geometrical features of the external surface as normal vectors and a set of points that can represent the surface in an organized way. Based on that information, a spray-painting path for a 6-DOF robot is then generated. The significant steps in this proposed methodology are presented in Figure 1. A common autonomous robotic spray-painting configuration is shown in Figure 2.

### 3.1. Acquisition of the 3D Point Cloud

When working with the 3D point cloud data-based path planning for autonomous spraying implementation, the critical challenge is the model reconstruction of 3D scanned workpieces. The 3D point cloud model of the workpiece can be obtained in two ways: either directly from a 3D scanning technique or by transforming an existing 3D CAD model into an STL format, as depicted in Figure 3, and then to point cloud format. In the present approach, functions from the open3D library (version 0.15.0) [19] are used to read point cloud data from different file formats such as ply, pcd, xyz, etc. Alternatively, a uniform sampling algorithm can transform the STL format into a 3D point cloud. The Poisson disk sampling method [20] is used in the present case because it can evenly distribute the points on the mesh surface. Another feature of this algorithm is that the number of sample points can be defined, as well as the minimum distance between two neighboring points. The more sample points used, the more accurate the 3D representation of the object will be, but a longer computation time is required to process the data.

### 3.2. Point Cloud Preprocesing

When the point cloud is extracted based on a CAD model using a sampling method, the data quality is acceptable for this application after slightly tuning the number of points. However, when data are collected from 3D scanning devices, it is common for the point cloud to contain noisy or unnecessary information that should be removed. The steps to prepare the point cloud are presented in Figure 3, depending on the source of the point cloud data. First, remove unnecessary sections such as fixtures, floors, or other scenes which were unavoidably recorded during the scanning process. Second, remove points at which the number of neighbors is less than a threshold inside a given sphere radius around them to eliminate sampling noise. Third, perform a point cloud down sampling; in this case, 10,000–20,000 points are shown to be enough to represent a workpiece properly. If the number of points exceeds that, a uniform sampling using the Poison disk algorithm is performed. The last step is to adjust the position and orientation of the model according to the actual pose during the spraying process, considering that the object can be hung or mounted on a particular fixture.

### 3.3. Process of 3D Data Extraction

A 3D point cloud lists unorganized points in 3D space without constructed geometrical information. An STL file provides, in addition, a connectivity list between the points to represent the surface of the object as a set of surfaces, but it is still an unorganized set of data. This section aims first to obtain the point connectivity of the object’s surface, expressed by a connectivity wireframe, in an organized way. Once we know the position of each point with respect to others, it is simple to extract data as normal vectors and calculate the spray-painting path. It is also easy to create a surface triangulation that can be used for other purposes, such as the paint quality simulation based on the exposure of the sprayed paint to each small area of the object’s surface.

#### 3.3.1. Use of 3D Principal Component Analysis (PCA)

Several methods can be used to define the orientation of a given point cloud. Among them, the axis-aligned bounding box (AABB) and oriented bounding box (OBB) are the most commonly used [21], and recent approaches include deep learning and iterative closest point (ICP) algorithms to determine the pose of an object [22]. For the purposes of this research in terms of simplicity, 3D principal component analysis (PCA) is a valuable tool to approximate the dominant data direction within entire data sets and is also the algorithm behind the calculation of an OBB for a given point cloud. In the case of point clouds, it helps to find a coordinate frame that represents the orientation of a model based on the dominant directions of the point cloud data. Following the work presented by [23,24], the process can be described as follows: Considering a 3D point set *P* = {(*x_i_*, *y_i_*, *z_i_*)} where *i* = 1, 2, 3, …, *n*, the means *x_m_*, *y_m_*, and *z_m_* are given by:(1)$$x_{m}=\frac{1}{n}\sum_{i=1}^{n}x_{i},\;{\;\;\;\;y}_{m}=\frac{1}{n}\sum_{i=1}^{n}y_{i},\;{\;\;\;\;z}_{m}=\frac{1}{n}\sum_{i=1}^{n}z_{i}$$
where *n* is the number of points in the 3D model. By subtracting the mean from each data point, the data variance values can be computed as:(2)$${\sigma_{xx}}^{2}=\frac{1}{n-1}\sum_{i=1}^{n}{\left(x_{i}-x_{m}\right)}^{2},\;\;\;\;\;$$
(3)$${\sigma_{xy}}^{2}=\frac{1}{n-1}\sum_{i=1}^{n}\left(x_{i}-x_{m}\right)\left(y_{i}-y_{m}\right),\;etc.\;$$
where ${\sigma_{xx}}^{2}$ is the variance of *x* and ${\sigma_{xy}}^{2},$ the cross covariance between the *x* and *y* variables. This gives as a result a 3-by-3 square matrix *C*:(4)$$C=\left[\begin{array}{ccc} {\sigma_{xx}}^{2} & {\sigma_{xy}}^{2} & {\sigma_{xz}}^{2}\\ {\sigma_{yx}}^{2} & {\sigma_{yy}}^{2} & {\sigma_{yz}}^{2}\\ {\sigma_{zx}}^{2} & {\sigma_{zy}}^{2} & {\sigma_{zz}}^{2} \end{array}\right]$$

The eigenvectors and eigenvalues of the covariance matrix *C* can be calculated by using the matrix diagonalization method, that is:(5)$$V^{-1}CV=D$$
where *D* is a diagonal matrix containing the eigenvalues $\left\{\lambda_{1},\lambda_{2},\lambda_{3}\right\}$ of *C*, and *V* is an orthogonal matrix that contains the corresponding eigenvectors. Because the covariance matrix is a symmetric semi-positive definite matrix, the obtained eigenvalues should be greater than or equal to zero, that is, $\lambda_{1}\geq \lambda_{2}\geq \lambda_{3}\geq 0$. Figure 4 shows the results of applying PCA to the point cloud of a metallic part. The coordinate axes represent the covariance matrix principal directions. The lengths of these axes correspond to the eigenvalues of the matrix, and their directions are defined by the directions of their eigenvectors. The eigenvector with the bigger eigenvalue is where the points have more variation. The eigenvectors give valuable information to create the sphere S to create the path on the target surface, as explained in the next section.

#### 3.3.2. Spherical Mesh Grid Generation

The most commonly used bounding volumes to “wrap” or encapsulate a point cloud are AABB, oriented bounding boxes OBB, and spheres [21], with most research focusing on applications such as collision avoidance and 3D reconstruction. In the current case, the sphere is the most generalized shape because its equation only requires a radius and the center coordinates, making computation easier in a variety of applications. For this application, a sphere is more advantageous than AABB and OBB because it surrounds an object on all sides and is effective at encapsulating both irregular and basic shapes. AABB and OBB are more suitable and limited to basic shapes with a “box-like” geometry.

A sphere is created to “wrap” the working part, which is an organized set of points and its triangulation that is meant to suffer a deformation to obtain a surface shape. A spherical mesh grid is shown to have better results when approximating complex surfaces by obtaining more information about the curvature than AABB or OBB, with the challenge of finding a proper position for the center of the sphere. Firstly, the mesh grid is created based on these parameters: sphere center *S_c_*, sphere radius *S_r_*, and sphere resolution *S_deg_*, which defines the angular separation between points (Figure 5a,b), with a smaller *S_deg_* value, but the number of points on the sphere *S* increases and the computation time also increases. The data on this sphere are represented by a set of vertexes *S* = {(*x_i_*, *y_i_*, *z_i_*)} where *i* = 1, 2, 3…*m*, and *m* is the number of points in the sphere. A set of normal vectors corresponding to each vertex, *S_N_* = {(*Nx_i_*, *Ny_i_*, *Nz_i_*)} where *i* = 1, 2, 3…*m*, a triangulation between these points is represented by *S_T_* = {(*p*_1*i*_, *p*_2*i*_, *p*_3*i*_)} where *i* = 1, 2, 3…*n*, and *n* is the number of triangles representing the surface, and *p*_1*i*_, *p*_2*i*_, *p*_3*i*_ are the points in *S* that connected in the order [*p*_1*i*_, *p*_2*i*_, *p*_3*i*_, *p*_1*i*_] forms the triangle *i*. Figure 5a,b show the spherical mesh grid with different resolutions, while in Figure 6 a section of the triangulation is represented, including points, normal vectors, and triangles.

The center of the wrapping sphere must be found depending on the part position and the target surface to paint. As the center of the part and the eigenvectors have been calculated, the center of the sphere can be defined by moving the center of the part an offset distance *d_offset_* in the direction of the eigenvector that is normal and moves in the opposite direction to the target surface. A graphical representation is presented in Figure 7a. When *d_offset_* is equal to 0, the center of the sphere and the center of the work piece are similar, and the radius of the sphere will be small. As the value of *d_offset_* increases, the radius of the sphere increases. The effect of wrapping a work piece in a small or big sphere can have different outputs depending on the approximation of the target surface. After several experiments, it was found that *d_offset_* can be approximated in the range of two times the maximum distance between the center of the part and the farthest point on the workpiece point cloud. When *d_offset_* tends to be infinite, the sphere section touching the surface of the work piece can be considered as wrapping with a plane.

#### 3.3.3. Wrapping Method for Convex and Concave Surfaces

After defining the sphere *S* and its center *S_c_*_,_ as shown in Figure 5c,d, the next step is to “wrap” the current point cloud by using the sphere *S*. The method consists of an iterative operation of decreasing the sphere radius by a given distance ∆d on each iteration until it reaches the center *S_c_*. At the same time, for each point on the sphere *S*, a neighbor search is performed to find any point on the point cloud *P* within a given search radius *r_s_*. When a point is found, the point on the sphere is projected in the point cloud *P* and stops moving towards the center of the sphere *S*. In Figure 7b, a graphical representation of the algorithm is presented. As the point cloud geometry might present features that allow for finding points on the inner surface of the part, as shown in Figure 8a, a stopping condition is added to the algorithm to prevent this from happening. In this case, a point *p_ij_* in *S* moving towards the center of the sphere will be stopped after one of the four adjacent neighbor points finds a point in point cloud *P*. In Figure 8a, the red line represents points that stopped after finding a point in the point cloud *P* without any stopping condition. In Figure 8b, the red line represents points that stopped after finding a point in the point cloud *P* or when reaching a given number of iterations after one of the adjacent points had been stopped.

Once the surface wrapping is complete, the normal vector *S_N_* must be updated to represent the proper direction of the local surface section. At the same time, a normal vector is calculated by obtaining the average of the normal vector of each line that connects a point *p_ij_* with its adjacent neighbors in all directions. Figure 7c shows the normal between two points in black dotted lines, and the average normal at that point is shown in green. The “wrapping algorithm” by contracting the sphere is helpful for convex surfaces where an exterior surface is of interest.

**Figure 7 sensors-23-09634-f007:**
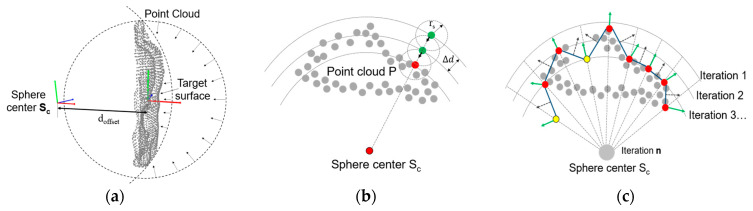
(**a**) Sphere center based on the robot position and workpiece principal components and spheres with centers in the centroid of the part and by applying an offset; (**b**) sphere decreasing diameter to find the point cloud surface; (**c**) sphere points after performing the wrapping method. The red points are fixed after touching the point cloud; the yellow points stop when the predefined number of iterations is reached after the neighboring point stops.

**Figure 8 sensors-23-09634-f008:**
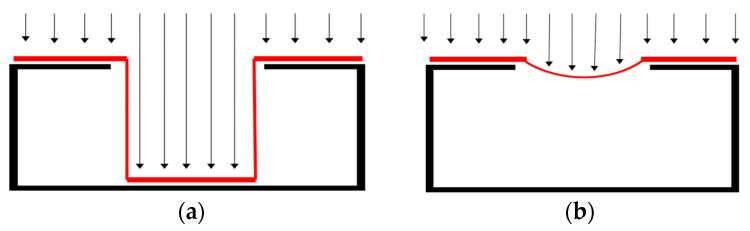
Example of how the wrapping algorithm works on planar surfaces: (**a**) the inner surface of the object is considered; (**b**) a parameter stops the wrapping algorithm from touching the inner surfaces.

#### 3.3.4. Surface Extraction and Preprocessing

As described in Section 3.3.3, a surface triangulation *S_T_* is already defined to have an organized set of data; however, after performing the wrapping method, only the points on sphere *S* that touch the point cloud *P* are of interest; as a consequence, the rest of the points must be ignored. It is easy to separate by adding an identifier or flag to each point that stops at some points during the process, and points that go all the way to the center of the sphere are ignored. Figure 9a,b present the wrapping algorithm after contraction, and the surface of interest is isolated. In the same way, a growing sphere can extract information from concave or inner surfaces, as shown in Figure 9c,d.

It is essential to mention that the path-planning system is intended to work directly on the 3D point cloud, so only the set of a vertex in the sphere *S* and its corresponding normal vectors are of interest for this process. The corresponding triangulation *S_T_* is useful for paint deposition simulation and paint quality validation after the trajectory is created.

## 4. Paint Deposition Model of Free-Form Surface

The geometric shape of free-form surfaces and the sophistication of spray patterns generated by current paint atomizers make this challenging. In this work, the paint deposition is modeled as conical, and the spray pattern is circular when the spray gun is perpendicular to a planar surface, as presented by [6] and shown in Figure 10a, where *r* is the distance of any point *p* in the point cloud *P* within the spray circle to the center of it; *φ* is the angle between the line formed from the spray nozzle to the point *p* and the center line of the conus; *h* and *R* are defined in Table 1.

The coating thickness delivery model is essential for spray profile preparation and efficiency optimization. Earlier researchers implemented mathematical methods to improve the spraying simulations, including the Cauchy distribution mechanism of infinite-range models [25], the Gaussian model [26], the finite-range models which are the parabolic distribution model [27], *β*-distribution model [28], the ellipse dual-*β* distribution model [29], and others. All these models are based on curve fitting that is designed for specific spray guns. However, experimental methods offer higher accuracy in the approximation for this specific task. A 2D deposition model represented by a Gaussian deposition model is built where the concept is approximated using a collection of data consisting of thickness measurements on all spots sprayed on a planar sheet [30], which is supplemented by using a similar method and approximation of the Gaussian parameters by using the Bayesian structured algorithm and the genetic algorithm [31].

For this research, the *β*-distribution model is used as it offers more flexibility to represent different patterns, including parabolic distribution (*β* = 1.5) and Gaussian distribution (*β* = 8), by changing the value of the parameter *β* as shown in Figure 10b. A colormap representation of different *β* values is represented in Figure 10c–f.

The spray-paint parameters needed to create a trajectory can be divided into two groups: parameters defined by the user, or inputs; and parameters that need to be calculated, or output parameters. Table 1 shows the list of parameters. The outputs parameters are calculated as follows:(6)R=h tan(θ)

The overlap distance *d* of two adjacent passes is a crucial parameter to be managed, since the paint distribution on a planar surface is not uniform. The overlap distance helps to compensate for this difference by considering two paint passes on specific sections of the surface along the path, as shown in Figure 11. Considering the *β*-distributed model to represent the paint thickness, a clear and more practical estimate of the overlap distance is feasible instead of approximating the paint distribution to parabolic or Gaussian.

The coating deposition diffusion model is influenced by several conditions to optimize the deposition process; typically, the variables such as temperature, air pressure, and humidity remain unchanged, as well as parameters such as the opening angle of the spray gun *θ* and the paint flux. Therefore, the paint thickness at a given point *p* can be represented as a *β*-distributed model as described below:(7)$$q(r)=\int_{0}^{t}{\frac{\eta Q_{0}\beta }{\pi R^{2}}\left(1-\frac{r^{2}}{R^{2}}\right)}^{\beta -1}dt\;for-R\leq r\leq R\;$$
(8)$$\dot{q}\left(r\right)={\frac{4\eta Q_{0}\beta }{\pi {vR}^{2\beta }}\left(\frac{{\left(R^{2}-r^{2}\right)}^{\beta -0.5}}{2\beta -1}\right)}^{\beta -1}for-R\leq r\leq R$$

The optimum paint coverage is determined from two different paint passes, *q*_1_(*r*) and *q*_2_(*r*), as the paint deposited by Paint Passes 1 and 2, respectively. As seen in Figure 11, the common region between the two paint passes is defined by the blue area where the two paint passes deposit the paint. The thickness of the paint can therefore be modeled as a piecewise function [14] and represented by the equations:(9)$$f(r,d)=\left\{\begin{array}{c} q_{1}\left(r\right),\;\;\;\;\;\;\;\;\;\;\;\;\;-R\leq r\leq R-d\\ q_{1}\left(r\right)+q_{2}\left(r\right),\;\;\;\;\;R-d\leq r\leq R\\ q_{2}\left(r\right)\;\;\;\;\;\;\;\;\;\;\;\;\;\;\;-R\leq r\leq 3R-d \end{array}\right.$$
(10)$$q_{1}(r)=\int_{0}^{t_{1}}f\left(r_{1}\right)dt\;\;\;\;\;\;\;\;\;\;-R\leq r_{1}\leq R$$
(11)$$q_{2}(r)=\int_{0}^{t_{2}}f\left(r_{2}\right)dt\;\;\;\;\;\;\;\;\;\;R-d\leq r_{2}\leq 3R-d$$
(12)$$t_{1}=\frac{\sqrt{{(R}^{2}-r^{2}})}{v},\;\;t_{2}=\frac{\sqrt{{(R}^{2}-(2R-d-{r)}^{2}})}{v}$$

$q_{1}\left(r\right)$ and $q_{2}(r)$ are given by Equation (8), $t_{1}$ and $t_{2}$ are the time of spraying on each paint pass, respectively. The overlap distance is calculated using the Min–Max optimization algorithm proposed by [30] to perform the task quickly and straightforwardly. The flowchart for the calculation is presented in Figure 12. The overlap distance *d* can be calculated independently from the spray-gun velocity *v* giving a value of one in this step.

Spray-gun velocity is considered as an input; once the overlap distance has been obtained, the velocity for a planar surface can be recalculated as presented by [27]:(13)$$v_{opt}=\frac{(\frac{1}{2}R-d)\int_{2R-d}^{0}V^{2}\left(r,d\right)dr+{V_{max}}^{2}\left(d\right)+{V_{min}}^{2}\left(d\right)}{q_{d}((\frac{1}{2}R-d)\int_{2R-d}^{0}V^{2}\left(r,d\right)dr+V_{max}\left(d\right)+V_{min}\left(d\right))}$$
where $V\left(r,d\right)=\frac{v\dot{q}\left(r\right)}{\eta }$, and $\dot{q}\left(r\right)$ is given by Equation (8).

When the surface is not planar, further calculations are needed to obtain the velocity, which must be recalculated after the path has been created to evaluate the uniformity of the paint applied to the surface. Further details are presented in Section 5.

## 5. Path-Planning Algorithm

Material deposition can be applied in many ways. However, for two main material deposition patterns, the tracer method is considered one of the best options [3]; consequently, for continuous material deposition, the raster and spiral patterns are commonly used in most material deposition processes. Given the nature of the target surfaces, the raster pattern is used in this research leaving other options for future research. The next step is autonomous path generation, in which the surface’s complexity determines how robust the algorithm should be.

Given the extracted surface as a section of the sphere *S*, in an organized way, first we select the starting points as a row or column of that array to be the first paint pass, and then a new paint pass is needed to be equidistant from the first pass. More points are formed by interpolating the position along each row or column and its related normal vectors to improve the algorithm’s accuracy. Then, the cumulative distance from the starting point is calculated until the cumulative distance is equal to or greater than *2R-d*, as shown in Figure 13a. At this moment, this point will be used in the following paint pass, and this process is repeated until it reaches the last point on the surface. Once the process is conducted for all the rows or columns, the passes are organized in a raster pattern to generate the path. Figure 13b shows a graphical description of the process.

In this section, each paint pass path is evaluated to remove noisy points, representing a change in the position and orientation of the tool outside a given boundary. After that, the path is smoothed using spline approximation, generating a more suitable trajectory for robot operation.

## 6. Robot Velocity Calculation

When the surface is not planar, the uniformity of the paint applied on the surface can be evaluated based on the selection of the optimal robot end effector velocity *v*. A multi-objective optimization algorithm based on a genetic algorithm (GA) is used. The global average paint thickness differs from the desired *q_d_* in this case. Therefore, the global velocity optimization value is formulated using a weighted-sum approach proposed by [32]. The paint thickness model for a point cloud *P* representing a surface with *n* points can be described in [6] by Equations (14) and (15) with a graphical representation in Figure 14.
(14)$$q_{p}=\left\{\begin{array}{c} \sum_{n=2}^{P_{path}}\frac{m_{n}}{v_{opt}}{\left(\frac{h}{l_{p}}\right)}^{2}\frac{\cos\left(\gamma_{p}\right)}{{cos}^{3}\left(\theta_{p}\right)}\dot{q}\left(r\right)\;\;\;\gamma_{p}\leq 90{}^{\circ}\;\;\;\;\;\;\;\\ 0\;\;\;\;\;\;\;\;\;\;\;\;\;\;\;\;\;\;\;\;\;\;\;\;\;\;\;\;\;\;\;\;\;\;\;\;\;\;\;\;\;\;\;\;\gamma_{p}\geq 90{}^{\circ}\;\; \end{array}\right.$$
(15)$${\dot{q}\left(r\right)=\frac{\;\eta Q_{0}\beta }{\pi R^{2}}\left(1-\frac{r^{2}}{R^{2}}\right)}^{\beta -1}$$
where $q_{t}$ is the paint thickness at point p in the point cloud *P*; $m_{n}$ is the Euclidian distance between the current path point and the next; $l_{p}$ is the distance between the spray-gun tip and point *p*; $\gamma_{p}$ is the angle formed by the average normal of the surface section and the spray-gun normal. $\theta_{p}$ is the angle that the normal vector at point *p* makes with the vector that points from the surface point *p* to the spray-gun center; $v_{opt}$ is the optimal velocity for the paint pass; $\dot{q}\left(r\right)$ is given by Equation (15). The total paint thickness at each triangle is then calculated by adding all the paint thickness values from Equation (14) based on the number of times a single point is sprayed [32]. Then, the optimization problem can be formulated as follows:(16)$$\min F=\left(F_{p},A\right)\;F_{p}=\left(\left|q_{t}-q_{d}\right|\right)\;and\;\;A=P-P_{qd}$$
where $q_{p}$ is given by Equation (14), $q_{d}$ is the desired thickness, *P* is the total number of points on the surface, and $P_{qd}$ is the total number of points with a paint thickness within the allowed variation ∆$q_{d}=q_{d}\pm q_{s}$. Here, *A* is included because minimizing $F_{p}$ is not enough; the number of points within the range ∆$q_{d}$ must be maximized. From Equation (9), the objective functions $F_{p}$ and *A* are conflicting. Also, for a surface represented with a total of *P* points, the number of equations to be solved is *P* + 1. To reduce the number of equations, the authors proposed converting the function $F_{p}$ into a single function *W* using the weighted sum approach [32], which is given by:(17)$$\min F=\left(W,A\right)\;\;\;for\;W=\sum_{p=1}^{P}w_{p}F_{p}$$

To achieve a practical solution for minimizing *F*, the weight $w_{p}$ for each value of $F_{p}$ is considered to be one, as we need to know which points are out of the allowed thickness range and this simplifies the proposed optimization. This optimization problem is solved using the MATLAB R2022a multi-objective genetic algorithm method *gamultiobj* to obtain a Pareto set of solutions. Once the Pareto set is calculated, choosing a unique optimal solution is necessary. For that, the next step is to minimize *W* and *A* independently by using the MATLAB R2022a genetic algorithm method *ga* to obtain the utopia point which is the intersection between the optimal solutions of W and A, each used as the single objective function alone, the optimal solution is the closest point on the pareto front to the utopia point, and can be obtained by calculating the Euclidean distance to all the optimal solutions and then finding the one corresponding to the minimum distance.

## 7. Variable Paint Flux Calculation

After the trajectory and robot end effector velocity have been calculated for a constant paint flux, the thickness deposition on the surface provides critical information about the trajectory’s performance in achieving thickness uniformity across the surface. To ensure a required level of quality, upper and lower limits must be set when evaluating thickness uniformity. According to The Society for Protective Coatings Paint Application Standard (SSPC) [33], a specific procedure should be followed to determine conformance to the dry coating thickness requirements. This procedure typically involves using a dry film thickness gauge to measure the coating thickness at various points on the surface and then comparing the measurements to the specified limits. The upper and lower limits for coating thickness are typically calculated based on the desired thickness and are defined according to a specified level ranging from 1 to 5. A level of 1 is more restrictive than other levels, while the default level is 3, which allows for a deviation of ±20% from the desired thickness. Level 4 allows for a deviation of −20% to +50%, while Level 5 allows for a deviation of −20% on the lower limit and no upper limit. By evaluating the thickness variation, it is possible to assess the performance of the trajectory. The paint flux along the trajectory can be modified if necessary while keeping the path poses and robot velocity constant. It contributes to the improvement of the trajectory’s performance. Figure 15 depicts the total approach for obtaining the final trajectory with changing paint flux.

The paint flux is changed to improve the thickness uniformity by identifying surface sections with under-painted or overpainted regions. The trajectory points that affect these areas are then located. For each point, the thickness of the painted surface is evaluated. The paint flux is then increased or decreased according to Equation (15) to maximize the number of small triangles on the surface that satisfy the required thickness limits. After calculating the paint flux for each trajectory point, a curve smoothing process is necessary to avoid abrupt changes. In this case, the Savitzky–Golay filter is used because it provides an acceptable balance between signal smoothing and preserving its features, minimizing the risk of signal distortion. Figure 16 in the simulation results section shows different plots of variable paint flux for different surfaces.

## 8. Simulation Results

To demonstrate the robustness and effectiveness of the proposed approach, four test cases have been set up, including a planar surface, a convex surface, a motorcycle engine cover, and a car clutch shell with highly complex geometry as shown in Figure 17.

More detailed information on these cases, including the surface coverage and thickness variation achieved by the algorithm, is shown in the following tables. Table 2 lists the input parameters considered for all the cases, while Table 3 presents the summarized output data for each case. The data are processed in a user interface developed in Python, as shown in Figure 18.

The feasibility of the trajectory in terms of robot motion must be validated to avoid singularities or undesirable robot movements. One research study [34] is specifically dedicated to this topic due to its complexity, and claims that it must be tailored to each type of robot. The literature [34] also presents an algorithm focused on controlling and limiting the speed and acceleration of the robot. However, in our research, we used the robot manipulation framework MoveIt 1 [35] and the Noetic Ninjemys version of robot operating system (ROS) [36] to calculate the robot’s inverse kinematics before running the trajectory on a real robot. After loading the generated trajectory into the simulation software, it might go through some changes as a result of the path-planning algorithm used and the robotic arm’s capacity to reach certain poses; in this case, the MoveIt 1, *Pilz Industrial Motion Planner* package is used to generate the trajectory. The affections mainly occur in the spray-gun orientation, as illustrated in Figure 19, where the tool orientation is modified gradually before the next pose to achieve smooth robot motion. After validating the trajectory feasibility on a given robot model, a robot code is generated to perform the task. Figure 20 shows such a motion test in a real robot.

According to the simulation results presented in Table 3, the proposed approach achieved 100% surface coverage for the planar surface, with 96.08% of the thickness within the allowed range when using a constant paint flux and 98.01% when using a variable paint flux with the calculated spray-gun velocity. Similarly, 100% surface coverage was achieved for the convex surface, and the thickness within the allowed range increased from 93.18% with a constant paint flux to 97.45% with a variable paint flux. The approach achieved 94.20% of the thickness within the allowed range for the motorcycle engine cover with a constant paint flux. The performance improved to 96.63% with a variable paint flux. For the car clutch shell case with a constant paint flux, only 73.68% of the thickness was within the allowed range, while the performance improved to 81.33% with a variable paint flux; however, the surface coverage was still high at 96.65%. The proposed approach showed promising results in achieving thickness uniformity and surface coverage for various surfaces. From the histogram information in Figure 21, Figure 22 and Figure 23 options *e* and *f*, the thickness variation tends to be within the range in planar or more regular surfaces, while for more complex surfaces as in Figure 24, the number of areas underpainted or overpainted increases. Figure 25 depicts the underpainted and overpainted areas in the car clutch cover part, where it is possible to see that these variations depend on the surface complexity and are most likely to occur in sections with abrupt geometry changes, such as edges, cavities, or areas with high roughness, because each robot tool orientation is calculated as the average of the surface normal in the affected sections, and some areas end up receiving the paint in a non-optimal direction. Another reason is that some of the path poses have been altered to avoid singularities or undesirable robot movements. Furthermore, spray-gun velocity and spray flux have a direct impact on the coat; when attempting to improve film quality in some areas, the immediate surroundings may be affected as well.

## 9. Conclusions

This paper describes a novel method for autonomously generating spray-painting trajectories with a robotic arm. A point cloud representation of the part is all that is needed for path planning, while a 3D reconstruction is not required. The proposed path-planning system can generate trajectories for objects of complex geometries with total parts coverage while minimal user interactions are required. A user interface (GUI) is designed to create trajectories given the input of painting parameters and show a visualization of the trajectory generated and a colormap simulation of paint film quality. This path-generation system is suitable for use in a variety of conveying systems and does not require the use of special fixtures because it is not dependent on a predefined workpiece position and pose.

The overlap distance between paint passes is calculated based on a planar surface model, then a raster pattern path is generated autonomously based on the given input parameters. The spray gun’s optimal constant velocity value is calculated using the paint deposition model and paint deposition simulation. The kinematic feasibility of the path generated is evaluated and modified in a simulation environment for use in a robot arm. The simulation shows that the thickness variation is kept within the allowed variation. While the average thickness is close to the desired thickness, the variation is still considerable when the part’s complexity is high. The complexity of the geometry, the accuracy of the approximation of the spray-gun model, and the robot’s kinematic feasibility will directly affect the quality of the coating film. The experiments show promising results in terms of total coverage when generating a path subject to a given paint quality requirement.

Further tests on more samples with varied shapes will be needed to fully identify the presented method’s weaknesses. Painting experiments on real objects are needed to compare the simulated results. Further improvements are required to construct a completely autonomous robotic spray-painting system capable of handling the high-mix and low-volume production lines. Robot motion planning must be studied in more detail to minimize the effect of the robot motion constraints on the final paint thickness.

## Figures and Tables

**Figure 1 sensors-23-09634-f001:**
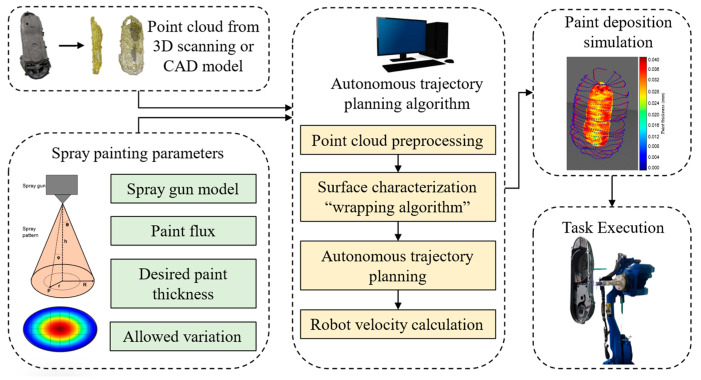
Autonomous spray-painting system workflow.

**Figure 2 sensors-23-09634-f002:**
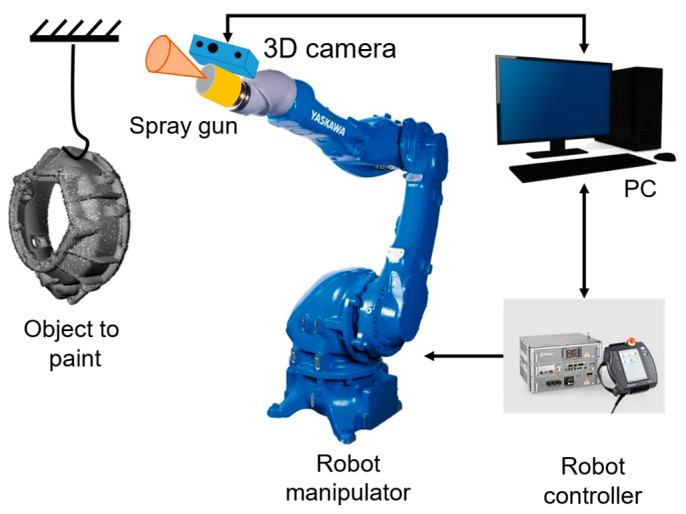
Autonomous spray-painting system architecture.

**Figure 3 sensors-23-09634-f003:**
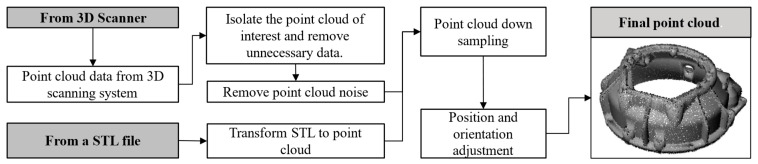
The 3D point cloud acquisition and preprocessing.

**Figure 4 sensors-23-09634-f004:**
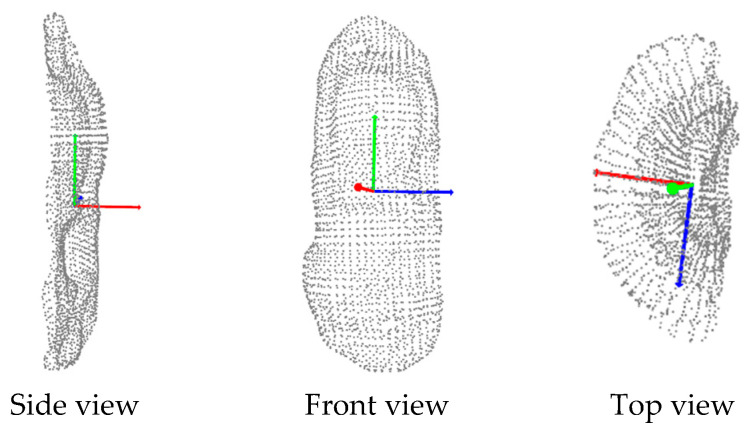
The 3D point cloud and the principal directions.

**Figure 5 sensors-23-09634-f005:**
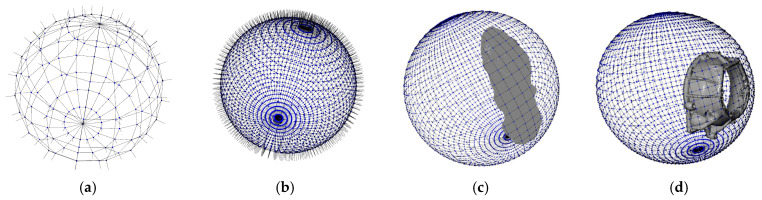
(**a**) Spherical mesh grid and normal vectors with *S_deg_* = 20°; (**b**) Spherical mesh grid and normal vectors with *S_deg_* = 5°; (**c**,**d**) Point cloud *P* and the spherical grid *S* for two metallic parts.

**Figure 6 sensors-23-09634-f006:**
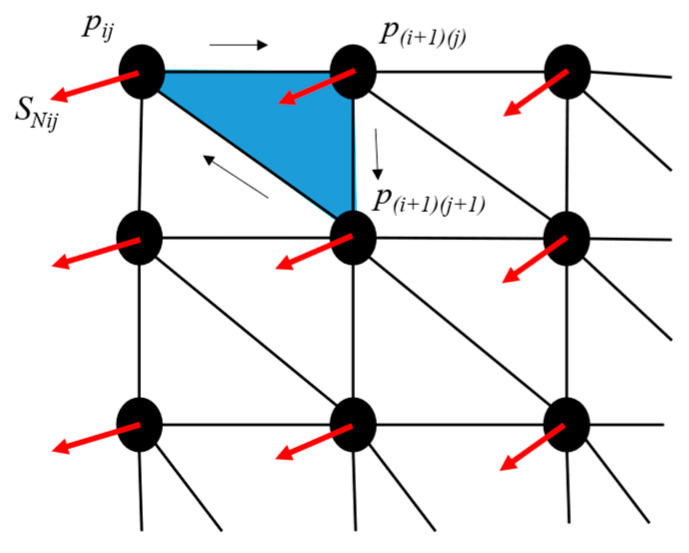
Representation of the data contained on sphere grid, points *p*, normal (*S_Nij_*), and triangle in blue.

**Figure 9 sensors-23-09634-f009:**
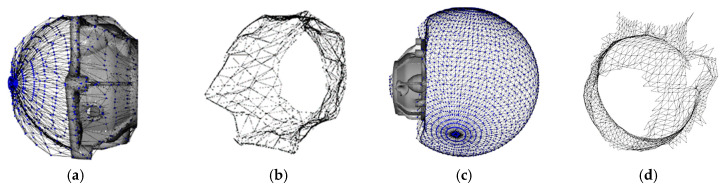
Surface extraction by “wrapping” the workpiece: (**a**) point cloud wrapping; (**b**) external surface of interest; (**c**) sphere growing to extract the inner surface of the workpiece; (**d**) internal surface of interest.

**Figure 10 sensors-23-09634-f010:**
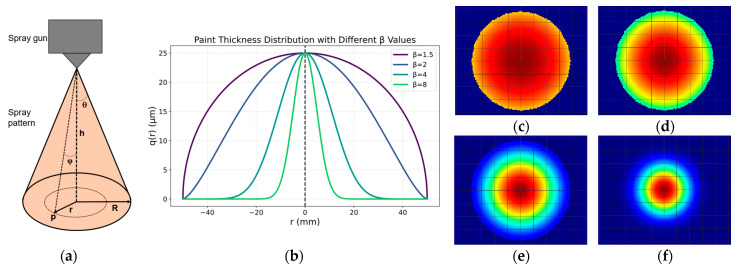
Paint deposition model on free form surfaces. (**a**) Spray-gun model with a circular pattern; (**b**) beta paint distribution model for varying *β*; and a circular paint deposition model on a planar surface, with radius *R* = 50 mm for: (**c**) *β* = 1.5; (**d**) *β* = 2; (**e**) *β* = 4, (**f**) *β* = 8.

**Figure 11 sensors-23-09634-f011:**
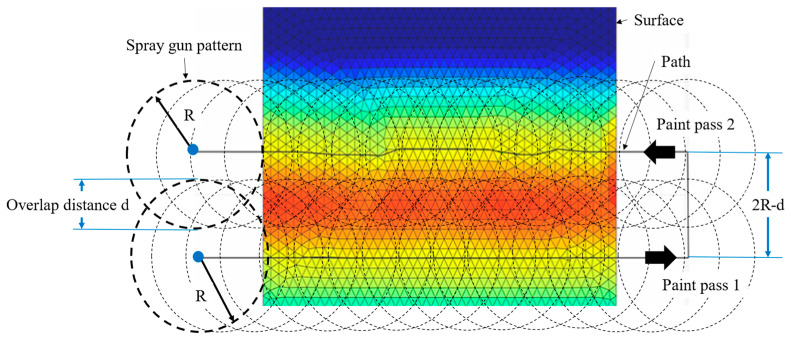
Overlap distance for a circular spray pattern on a planar surface.

**Figure 12 sensors-23-09634-f012:**
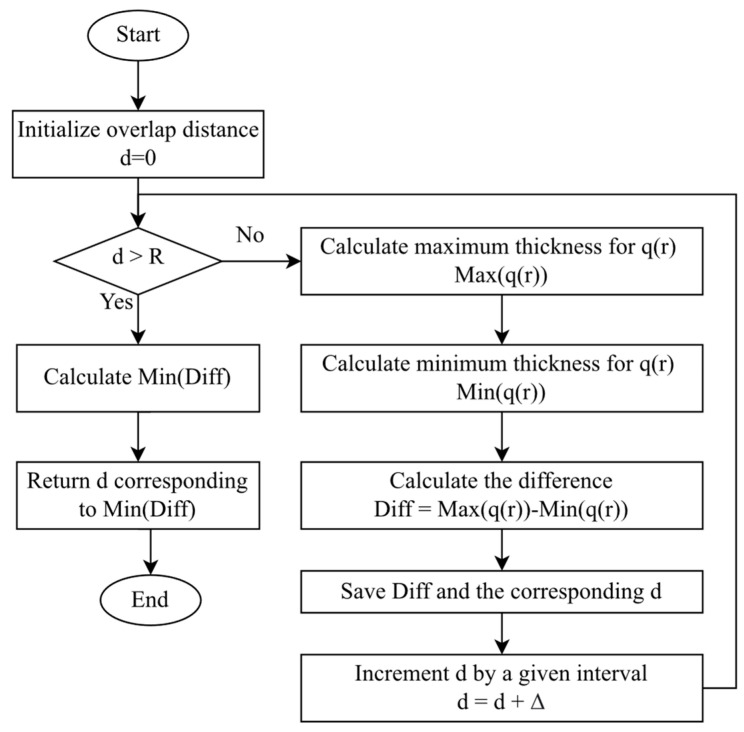
Flowchart of the calculation process to obtain the best overlap distance.

**Figure 13 sensors-23-09634-f013:**
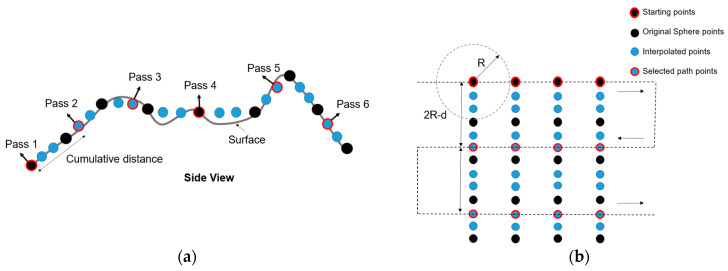
Path-planning algorithm based on point cloud: (**a**) front view of path planned; (**b**) path points extraction based on cumulative distance.

**Figure 14 sensors-23-09634-f014:**
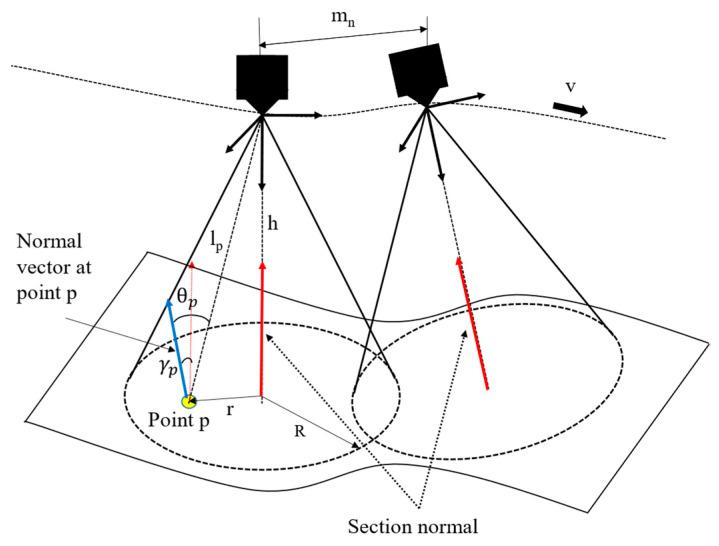
Free-form surface and projected spray-gun pattern.

**Figure 15 sensors-23-09634-f015:**
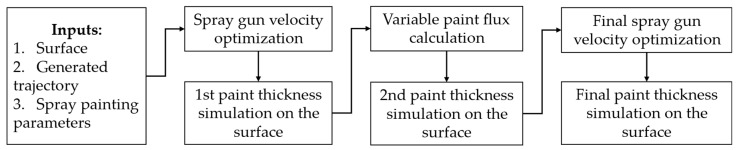
Overall procedure to evaluate a trajectory with variable paint flux.

**Figure 16 sensors-23-09634-f016:**
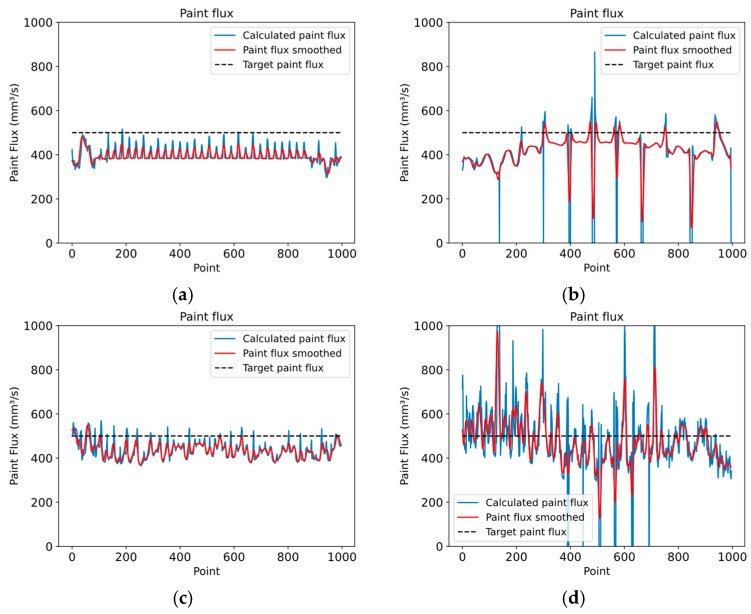
The plots show the calculated variable paint flux for (**a**) a planar surface; (**b**) a convex surface; (**c**) a motorcycle transmission cover; and (**d**) a car clutch shell.

**Figure 17 sensors-23-09634-f017:**
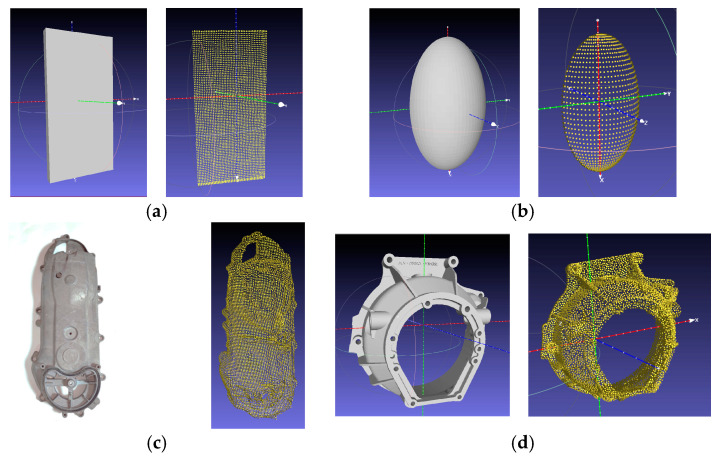
Sample geometries used for testing the proposed system based on a CAD model or from 3D scanning and the corresponding point cloud: (**a**) a planar surface; (**b**) a convex surface; (**c**) a motorcycle transmission cover; and (**d**) a car clutch shell.

**Figure 18 sensors-23-09634-f018:**
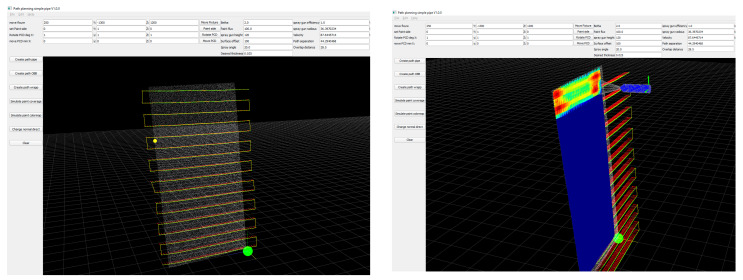
Python user interface to load point cloud, create and visualize the path, and simulate paint thickness.

**Figure 19 sensors-23-09634-f019:**
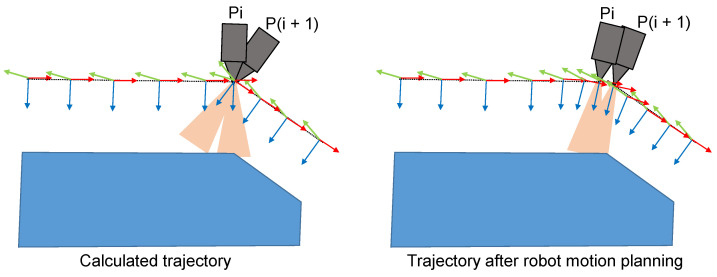
Motion-planning algorithm effect in the original generated trajectory.

**Figure 20 sensors-23-09634-f020:**
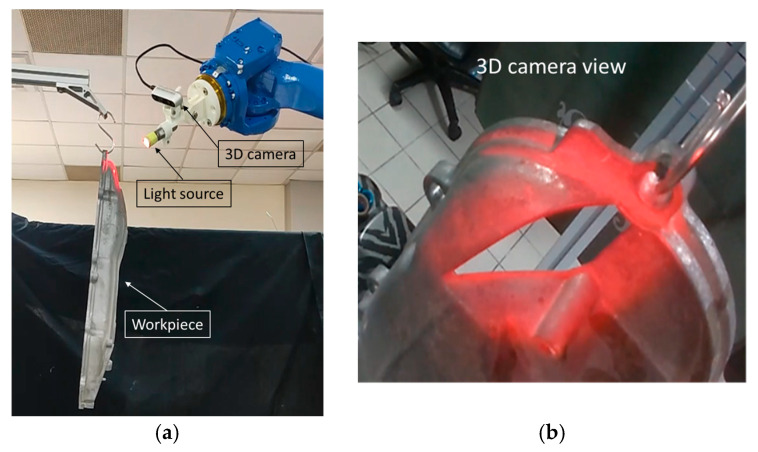
Total coverage trajectory validation in a 6-DOF robotic arm (Motoman GP50) using a light source simulating the spray gun. (**a**) System setup and (**b**) the light projected on the surface.

**Figure 21 sensors-23-09634-f021:**
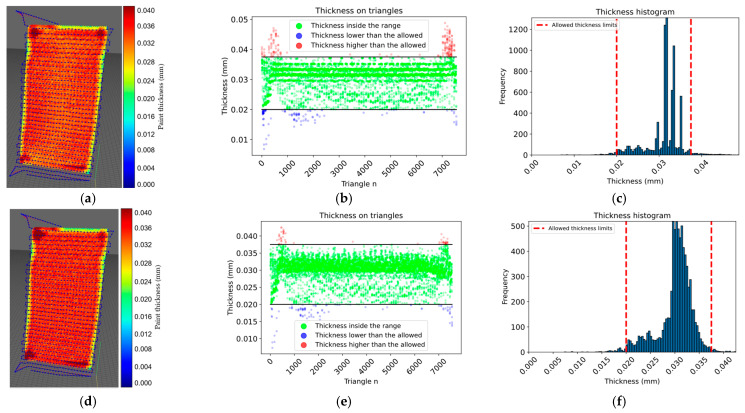
The images show the test results on a planar surface. In panels (**a**–**c**), the surface color map, thickness on each triangle, and a histogram representing the thickness distribution within the allowed range are shown, respectively, for a constant paint flux. In panels (**d**–**f**), the same parameters are shown, but for a variable paint flux.

**Figure 22 sensors-23-09634-f022:**
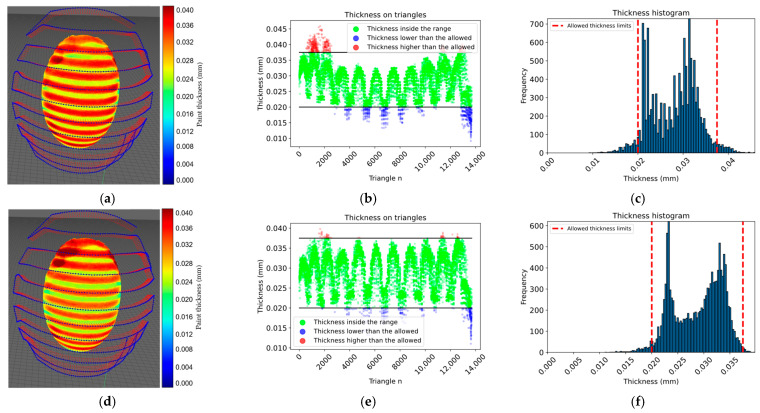
The images show the results of a test on a convex surface. In panels (**a**–**c**), the surface color map, thickness on each triangle, and a histogram representing the thickness distribution within the allowed range are shown, respectively, for a constant paint flux. In panels (**d**–**f**), the same parameters are shown, but for a variable paint flux.

**Figure 23 sensors-23-09634-f023:**
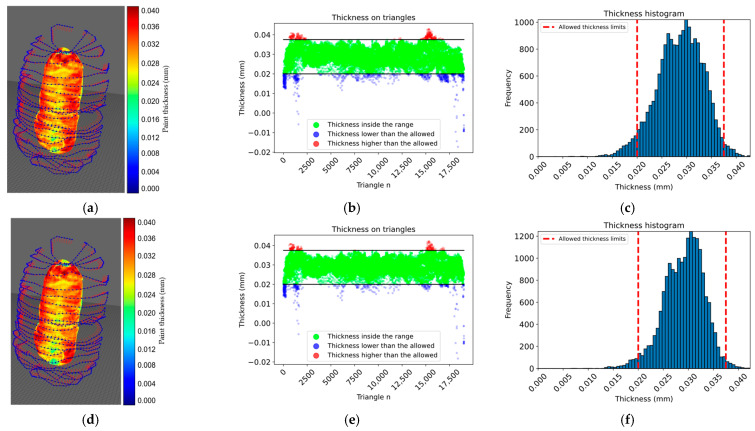
Shows the test results on a motorcycle transmission cover. In panels (**a**–**c**), the surface color map, thickness on each triangle, and a histogram representing the thickness distribution within the allowed range are shown, respectively, for a constant paint flux. In panels (**d**–**f**), the same parameters are shown, but for a variable paint flux.

**Figure 24 sensors-23-09634-f024:**
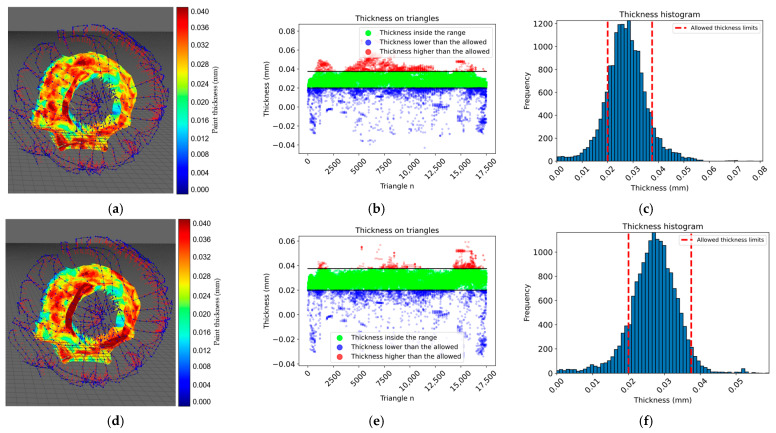
The images show the test results on a car clutch shell. In panels (**a**–**c**), the surface color map, thickness on each triangle, and a histogram representing the thickness distribution within the allowed range are shown, respectively, for a constant paint flux. In panels (**d**–**f**), the same parameters are shown, but for a variable paint flux.

**Figure 25 sensors-23-09634-f025:**
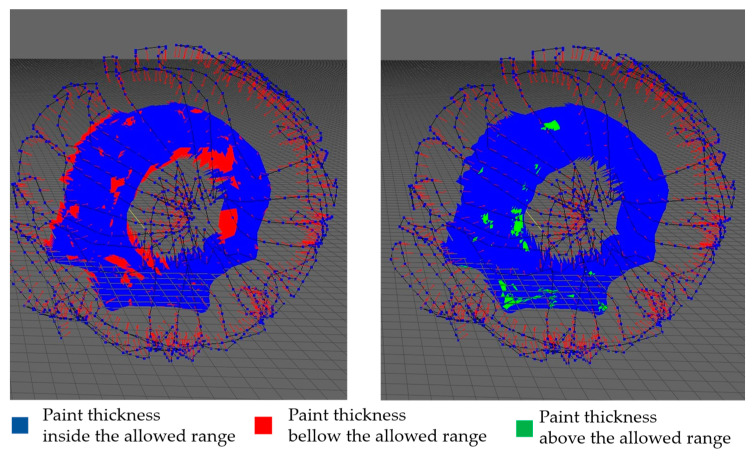
The images show the test results on a car clutch shell showing the overpainted areas (green) and underpainted areas (red).

**Table 1 sensors-23-09634-t001:** Input and output parameters for an autonomous spray-painting system.

Input Parameters	Output Parameters
Paint flux *Q*	Radius of spray circle *R*
Desired thickness *q_d_*	Overlap distance *d*
Allowed thickness variation *q_s_*	Spray-gun velocity *v*
Value *β*	
Spray-gun efficiency *η*	
Spray-gun angle *θ*	
Spray-gun standoff *h*	

**Table 2 sensors-23-09634-t002:** Input parameters used to test the system.

Input Parameters	Output Parameters
Paint flux *Q* (max)	1000 mm^3^/s
Desired thickness *q_d_*	0.025 mm
Allowed thickness variation *q_s_*	−20%/+50%
Value *β*	2
Spray-gun angle *θ*	20°
Spray-gun efficiency *η*	1
Spray-gun standoff *h*	100 mm

**Table 3 sensors-23-09634-t003:** Comparison table of the results of the simulation on different surfaces.

		Velocity (mm/s)	Average Thickness (mm)	Thickness within the Allowed Variation (%)			Surface Coverage (%)
				Bellow Range	Inside Range	Above Range	
Planar surface	Constant flux	246.72	0.0312	1.53	96.08	2.37	100
	Variable flux	255.49	0.0301	1.27	98.01	0.71	
Convex Surface	Constant flux	454.06	0.0280	4.04	93.18	2.77	100
	Variable flux	379.15	0.0289	2.16	97.45	0.37	
Motorbike engine cover	Constant flux	585.83	0.0285	4.12	94.20	1.67	99.98
	Variable flux	516.41	0.0290	2.37	96.63	0.98	
Transmission cover	Constant flux	523.60	0.0259	18.06	73.68	8.25	96.65
	Variable flux	551.72	0.0255	15.12	81.33	3.54	

## Data Availability

The data presented in this study are available on request from the corresponding author. The data are not publicly available due to privacy.

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
