# Peer review of "Autonomous Trajectory Planning for Spray Painting on Complex Surfaces Based on a Point Cloud Model"

_sensors, 2023, doi:10.3390/s23249634_

Round 1

Reviewer 1 Report

Comments and Suggestions for Authors

The paper has been written well overall.

A sphere was used to wrap the target object in the paper. Why don't you use an ellipse based on the dominant data direction, eigen vectors and eigen values of the target object?

In section 8 Simulation Results, more detailed analysis on the unsufficiently painted parts is needed with some examples. For example, what part couldn't be covered in painting frequently, or what part was painted with higher thickness than allowed, and so on. Then, what could be the possible reason for that. It could be some hints for modification of the method.

There are several typos.

Line 31: Whit 

Title of Table 2 and 3 are same.

Line 481: high complex geometry as shown in Error! Reference source not found..

Author Response

Firstly, we would like to thank the reviewer for his/her in-depth and careful reading of our manuscript and valuable suggestions. We have revised the manuscript by following the reviewer’s advices as much as possible. We have tried our best to provide all suitable answers for the questions according to our knowledge. Please see the attachment.

Reviewer 2 Report

Comments and Suggestions for Authors

The paper presents a method for the autonomous generation of the trajectory of a spray-painting robot based on a point cloud model. Although the paper is interesting and easy to read, I have several remarks to improve the overall quality of the manuscript.

1)      First of all, in my opinion, the main novelties of the work are not clear. There is not enough explanation of the originality of the proposed approach with respect to similar works on the topic of autonomous generation of trajectories in the field of spray painting.

2)      Second, the review of the literature review is poor and do not consider sufficient recently published works on the topic. Please see the suggested references below.

3)      The main results of the work are not clear. What are the main advantages of the proposed approach?

4)      It is not clear how the trajectories for the robot are planned in terms of motion law. Furthermore, please describe more in detail how the robot is controlled. Examples of robot trajectories (i.e., joint position, velocity and acceleration over time) should be reported in the paper.

5)      The quality of the figures is poor: in many graphs the size of the labels is too small and the resolution is not sufficient for a journal publication.

Suggested references:

·         Weber, A. M., Gambao, E., Brunete, A. (2023). A Survey on Autonomous Offline Path Generation for Robot-Assisted Spraying Applications. In Actuators (Vol. 12, No. 11, p. 403). MDPI.

·         McGovern, S., Xiao, J. (2023). A General Approach for Constrained Robotic Coverage Path Planning on 3D Freeform Surfaces. IEEE Transactions on Automation Science and Engineering.

·         Trigatti, G., Boscariol, P., Scalera, L., Pillan, D., Gasparetto, A. (2019). A look-ahead trajectory planning algorithm for spray painting robots with non-spherical wrists. In Mechanism Design for Robotics: Proceedings of the 4th IFToMM Symposium on Mechanism Design for Robotics (pp. 235-242). Springer International Publishing.

·         Wang, Z., Fan, J., Jing, F., Liu, Z., Tan, M. (2019). A pose estimation system based on deep neural network and ICP registration for robotic spray painting application. The International Journal of Advanced Manufacturing Technology, 104, 285-299.

·         Boscariol, P., Gasparetto, A., Scalera, L. (2023). Path Planning for Special Robotic Operations. In Robot Design: From Theory to Service Applications (pp. 69-95). Cham: Springer International Publishing.

·         Gleeson, D., Jakobsson, S., Salman, R., Ekstedt, F., Sandgren, N., Edelvik, F., ... Lennartson, B. (2022). Generating optimized trajectories for robotic spray painting. IEEE Transactions on Automation Science and Engineering, 19(3), 1380-1391.

Author Response

(The authors gave the same response as above.)

Round 2

Reviewer 2 Report

Comments and Suggestions for Authors

The paper was improved with respect to the previous version.